# Temperature-Dependent Biology and Population Performances of the Coffee Berry Borer *Hypothenemus hampei* (Ferrari) (Coleoptera: Curculionidae: Scolytinae) on Artificial Diet

**DOI:** 10.3390/insects14060499

**Published:** 2023-05-29

**Authors:** Shao-Hua Wei, Liang-Jong Wang, Ming-Ying Lin

**Affiliations:** 1Department of Plant Medicine, National Chiayi University, Chiayi 600355, Taiwan; edward831004@gmail.com; 2Division of Forest Protection, Taiwan Forestry Research Institute, Taipei 100051, Taiwan; ljwang23@ms17.hinet.net

**Keywords:** temperature, life history, survival rate, fecundity, longevity, population parameters

## Abstract

**Simple Summary:**

The coffee berry borer, *Hypothenemus hampei* is the most destructive pest of coffee in the world. It can damage the berries on coffee trees resulting in huge economic losses. Despite causing severe economic losses to the coffee industry worldwide, there have been few comprehensive studies on the life history and population performances of *H. hampei* due to its extraordinary ecological habits. Only a few studies have discussed its development, and there is no relevant research about the reproduction, longevity and population performances of this insect. In this study, we evaluated the survival rate and fecundity of *H. hampei* at different observation intervals. The observation intervals significantly affected the biological behavior of this pest. Moreover, we also compared the life history and population performances of *H. hampei* reared on an artificial diet at different temperatures. In summary, our study gathered and analyzed comprehensive biological information about *H. hampei*, which could be an important reference for future applied research, providing background knowledge useful for the management of this pest.

**Abstract:**

At different observation intervals of 1, 5, and 10 days during a trial period of 30 days, the mortality rates of *Hypothenemus hampei* were 100, 95, and 55%, and the fecundity rates were 0.55, 8.45, and 19.35 eggs/female, respectively. At temperatures of 18, 21, 24, and 27 °C, the development time of the immature stage of *H. hampei* was significantly shortened with increasing temperature. Furthermore, the lower developmental threshold (*T*_0_) and thermal summation (*K*) of the immature stage were 8.91 °C and 485.44 degree-days, respectively. The greatest longevity of female and male adults reached 115.77 and 26.50 days, respectively, at 18 °C. The highest fecundity was 29.00 eggs/female at 24 °C. The population parameters of *H. hampei* were analyzed on the basis of the age–stage, two-sex life table theory. According to the data, the parameters were significantly affected by temperature. The highest net reproductive rate (*R*_0_) was 13.32 eggs/individual at 24 °C. The highest intrinsic rate of increase (*r*) and finite rate of increase (*λ*) were calculated as 0.0401 and 1.0409 day^−1^, respectively, at a temperature of 27 °C. The shortest mean generation time (*T*) was 51.34 days at 27 °C. Overall, we provide a discussion on comprehensive biological information regarding *H. hampei*, thus providing basic knowledge for further research on this pest.

## 1. Introduction

*Coffea* (Rubiaceae) is a special crop for beverages. *Coffea arabica* and *Coffea canephora* are the most common species for commercial trade [1]. With a planting area of approximately 11 million hectares and an annual production of approximately 9 million metric tons worldwide, coffee is one of the most economically valuable agricultural products in international trade [2]. In Taiwan, the coffee-planting area is about 1153 hectares, with an annual production of about 970 metric tons [3]. In recent years, the demand for coffee has increased significantly with the rapid increase in consumer groups, and the planting area has also increased year on year. The most common coffee pests in the world include the coffee berry borer *Hypothenemus hampei* (Ferrari), the hemispherical scale *Saissetia coffeae* (Walker) (Hemiptera: Coccidae), the green scale *Coccus viridis* (Green) (Hemiptera: Coccidae), and the red borer *Zeuzera coffeae* Nietner (Lepidoptera: Cossidae) [4]. In Taiwan, the coffee berry borer is the most important pest with respect to coffee cultivation [5,6].

The coffee berry borer (CBB), *Hypothenemus hampei* (Coleoptera: Curculionidae: Scolytinae), is considered to be the most destructive pest of coffee worldwide [7,8,9,10,11]. It affects coffee production tremendously, causing an annual economic loss of over USD 500 million worldwide, and has thus garnered widespread attention as a major pest [2]. In 2007, *H. hampei* was first discovered in Dongshan District, Tainan City, Taiwan [5]. It has since spread across the whole island following the cultivation of coffee, seriously threatening the coffee industry in Taiwan.

Despite causing severe economic losses to the coffee industry worldwide, there are few published studies focusing on the life history of *H. hampei* due to its extraordinary ecological habits. Only a few studies in the literature discuss theimmature-stage development, reproduction, and population performance of this insect [12,13,14,15]. The performance of its development and reproduction under different relative humidities and artificial diets has been studied [16], in addition to the performance of its reproductive potential under continuous feeding with an artificial diet [12], and the life table under field conditions [13]. Additionally, there have been discussions on the rearing density and fecundity of females [14], and the development and prediction of field populations at different altitudes in Hawaii [15]. The development, longevity, reproductive potential, and population performance of insects are affected by environmental factors. In order to comprehensively understand the impact of different environmental factors on pests, it is necessary to perform life history experiments under laboratory conditions. In addition, the study of basic ecology is the primary way to understand these pests. For this purpose, the age–stage, two-sex life table provides precise parameters that comprehensively describe the development, survival, and reproduction of populations [17]. Because the female age-specific life table is not able to clearly describe the stage differentiation and the contribution of the male population, using the age–stage, two-sex life table makes it possible to more accurately and completely interpret the ecology of pest populations [18,19,20].

In order to grasp the complete information of the life history of *H. hampei*, an experiment was carried out with an artificial diet. This study discusses the biology of *H. hampei*, including its survival and fecundity, at different observation intervals as well as the life history and population performance at different temperatures. This study will allow us to establish a comprehensive body of ecological information on *H. hampei*, that can serve as a major reference for relevant biological research on this pest, enabling effective strategies to be developed for integrated pest management of *H. hampei*.

## 2. Materials and Methods

### 2.1. Collection and Rearing of H. hampei

*Hypothenemus hampei* were collected from an organic coffee farm (23°25′17.9′′ N, 120°31′55.7′′ E) in Chiayi, Taiwan, in October 2019. The larvae, pupae, and adults were moved to flat-bottomed glass tubes (2.5 cm in diameter, 12 cm in height) containing artificial diets. The glass tubes were kept at room temperature for those insect colonies with natural reproduction without interference, and covered with black fabric to create a dark environment. The artificial diet was prescribed with reference to Brun et al. (1993) [21] for continual rearing of *H. hampei*. The diets were placed in flat-bottomed glass tubes and Petri dishes (3 cm in diameter) for the insect colony and life history experiments, respectively.

### 2.2. Survival and Fecundity at Different Observation Intervals

The survival and fecundity of *H. hampei* were recorded at three different observation intervals, including 1 (OB1), 5 (OB5), and 10 days (OB10). For each treatment, the female adults (*n* = 20) were selected randomly from the insect colony and reared individually in Petri dishes (3 cm in diameter) with the artificial diet (0.5 × 0.5 × 0.5 cm^3^). Rearing containers and artificial diets were replaced regularly during the observation of survival and fecundity. The Petri dishes were kept in incubators at 27 ± 0.5 °C and 70% ± 5% RH with a photoperiod of 12:12 h (L:D), using black fabric for shading to simulate the dark environment within coffee berries. The total observation period of the experiment was 30 days; that is, treatments OB1, OB5, and OB10 were conducted 30, 6, and 3 times, respectively.

### 2.3. Temperature-Dependent Life History

Female adults (*n* = 300) were selected randomly from the insect colony and moved to 10 Petri dishes (3 cm in diameter) with a layer of artificial diet for laying eggs. The Petri dishes were kept in incubators at 27 ± 0.5 °C and 70% ± 5% RH with a photoperiod of 12:12 h (L:D), covered with black fabric. After 5 days, the eggs (*n* = 80–120) were taken out gently using a fine brush and placed on other Petri dishes (3 cm in diameter) with the artificial diet at different temperatures (18, 21, 24, 27, and 30 °C). The development of the eggs was observed and recorded daily under the microscope. After the larvae hatched, they were placed individually in Petri dishes (3 cm in diameter) with the artificial diet (0.5 × 0.5 × 0.5 cm^3^). Their survival was observed and recorded daily along with the development of larvae and pupae until the adults emerged. The artificial diet was replaced regularly to ensure its freshness and sufficiency.

After the emergence of the adults, they were paired according to their relative sexes from the insect colony. Each pair was reared in a Petri dish (3 cm in diameter) with the artificial diet (0.5 × 0.5 × 0.5 cm^3^) for feeding and reproduction. The fecundity and longevity of the adults were recorded every 5 days, and the artificial diet was also simultaneously replaced. The data regarding individuals removed from the insect colony for pairing were not considered in the life history study.

The life history study was conducted at temperatures of 18, 21, 24, 27, and 30 ± 0.5 °C, with 70% ± 5% RH and a photoperiod of 12:12 h (L:D). Black fabric was used to shield the test equipment from the light to simulate the dark environment. More than 80 individuals of *H. hampei* were observed at each temperature in this study.

### 2.4. Statistical Analysis

The fecundity data at different observation intervals and development times, and the longevity and fecundity parameters of *H. hampei* at different temperatures were processed via analysis of variance (ANOVA) using the general linear model (PROC GLM). When significant differences between treatments were observed (*p* < 0.05), Tukey’s honest significant difference (HSD) test was used [22]. In addition to the comparison of the development time, longevity, and fecundity parameters at different temperatures, the development time and longevity between different sexes were also considered in the life history study.

To estimate the lower developmental threshold (*T*_0_) and the thermal summation (*K*), the development rate (day^−1^) of *H. hampei* in each stage was analyzed under different temperatures using linear regression (PROC REG). The linear regression equation is as follows:$$r\left(T\right)=a+bT$$
where $r\left(T\right)$ is the development rate at the temperature *T*, and *a* and *b* are the intercept and slope of the linear regression model, respectively. The lower developmental threshold (*T*_0_) indicates the temperature at which the development rate is 0, and is expressed as follows:$$T_{0}=-a/b$$

The thermal summation (*K*) is calculated as follows:K=1/b

The temperature-dependent life history raw data were analyzed using the age–stage, two-sex life table program TWOSEX-MSChart [23] to clarify the population parameters of *H. hampei* at different temperatures [18,19]. The life table includes the development time, adult pre-oviposition period (APOP), total pre-oviposition period (TPOP), oviposition days (Ovi-days), longevity, and fecundity. The age–stage-specific survival rate (*s_xj_*), the age–stage-specific fecundity (*f_xj_*), the age-specific survival rate (*l_x_*), and the age-specific fecundity (*m_x_*) were calculated on the basis of the daily survival rate and the fecundity of each individual in the studied population. The age–stage-specific survival rate (*s_xj_*) represents the probability of each individual surviving to age *x* and stage *j*, which can be simplified as follows:$$s_{xj}=\frac{n_{xj}}{n_{01}}$$
where *n*_01_ indicates the initial number of individuals in the life history study and *n_xj_* refers to the number of individuals surviving to age *x* and stage *j*. The age-specific survival rate (*l_x_*) represents the probability of a newborn individual surviving to age *x*, and is calculated as follows:$$l_{x}=\sum_{j=1}^{\beta }s_{xj}$$
where *β* is the number of stages. The age–stage-specific fecundity (*f_xj_*) is the mean fecundity of individuals at age *x* and stage *j*. The age-specific fecundity (*m_x_*) is calculated as follows:$$m_{x}=\frac{\sum_{j=1}^{\beta }s_{xj}f_{xj}}{\sum_{j=1}^{\beta }s_{xj}}$$

The population parameters include the net reproductive rate (*R*_0_), the intrinsic rate of increase (*r*), the finite rate of increase (*λ*), and the mean generation time (*T*). The net reproductive rate (*R*_0_) refers to the mean number of offspring produced by an individual, which can be a female, a male, or an individual that died in the immature stage. The formula is as follows:$$R_{0}=\sum_{x=0}^{\infty }\sum_{j=1}^{\beta }s_{xj}f_{xj}=\sum_{x=0}^{\infty }l_{x}m_{x}$$

The intrinsic rate of increase (*r*) indicates that the population will increase at a rate of “*e^r^*” with infinite time, and the population will reach a stable age–stage distribution (SASD), which is expressed as follows:$$\sum_{x=0}^{\infty }e^{-r\left(x+1\right)}l_{x}m_{x}=1$$
where *x* is the age starting from 0 [24] with the time unit “day”. The finite rate of increase (*λ*) means that the population will increase at the rate “*λ*” with infinite time, and the population will reach a stable age–stage distribution (SASD), which is expressed as follows:$$\sum_{x=0}^{\infty }\lambda^{-\left(x+1\right)}l_{x}m_{x}=1$$

Thus, the relationship between *r* and *λ* is as follows:$$\lambda =e^{r}$$

The mean generation time (*T*) refers to the time required for a population to increase *R*_0_ times with infinite time, where the population will reach a stable age–stage distribution (SASD). To be more concise, the above concept can be described as shown below:$$\lambda^{T}=e^{rT}=R_{0}$$

Accordingly, the formula for the mean generation time (*T*) is as follows:$$T=\frac{\ln R_{0}}{r}$$

In addition, the age–stage-specific life expectancy (*e_xj_*) at age *x* and stage *j* is calculated as follows:$$e_{xj}=\sum_{i=x}^{\infty }\sum_{y=j}^{\beta }s_{iy}^{\prime }$$
where $s_{iy}^{\prime }$ is the probability of an individual at age *x* and stage *j* surviving to age *i* and stage *y*, and it is calculated on the premise that $s_{xj}=1$ [19,25]. In other words, $s_{iy}^{\prime }$ indicates the time span of survival expected for an individual. The age–stagespecific reproductive value (*v_xj_*) indicates the contribution made by an individual at age *x* and stage *j* to the future population [26,27], and is calculated as follows:$$v_{xj}=\frac{e^{r\left(x+1\right)}}{s_{xj}}\sum_{i=x}^{\infty }e^{-r\left(i+1\right)}\sum_{y=j}^{\beta }s_{iy}^{\prime }f_{iy}$$

Accordingly, if x=0, j=1 (i.e., the newborn egg), and $s_{01}=1$, then the formula will become the following:$$\sum_{i=0}^{\infty }e^{-r\left(i+1\right)}\sum_{y=1}^{\beta }s_{iy}^{\prime }f_{iy}=1$$

Moreover, the reproductive value ($v_{01}$) of the newborn egg is calculated as follows:$$v_{01}=\frac{e^{r\left(0+1\right)}}{s_{01}}\sum_{i=x}^{\infty }e^{-r\left(i+1\right)}\sum_{y=j}^{\beta }s_{iy}^{\prime }f_{iy}=e^{r}=\lambda$$

Therefore, the reproductive value ($v_{01}$) of an individual at age 0 and stage 1 (i.e., the newborn egg) will be $v_{01}=\lambda$, and its value of contribution to the future population is the finite rate of increase (*λ*).

The means and standard errors of the population parameters were estimated using the bootstrap resampling method (*B* = 100,000) [28,29,30]. Significant differences between treatments were processed through the paired bootstrap test using TWOSEX-MSChart [23].

## 3. Results

### 3.1. Survival and Fecundity at Different Observation Intervals

The survival rates and cumulative fecundity of *H. hampei* at different observation intervals are shown in Figure 1 and Table 1. The observation interval significantly affected the survival rate of *H. hampei*. During the study period of 30 days, the survival rates of OB1, OB5 and OB10 decreased by 100, 65, and 30%, respectively, on the 20th day. The curve of the survival rate for OB1 presented a sharp decline, and all individuals died on the 18th day (Figure 1). Specifically, the longer observation interval led to a higher survival rate. On the 20th day of the study period, the cumulative fecundities of OB1, OB5, and OB10 were 0.55, 7.95, and 14.60 eggs/female, respectively. The cumulative fecundity of female adults was also significantly affected by the different observation intervals (*p* < 0.0001). Furthermore, the cumulative fecundity of OB10 was 35.18 times that of OB1 on the 30th day. Similar to the survival rate, there was higher fecundity under the longer observation interval. Therefore, we conducted a follow-up life history study of adult longevity and female fecundity at 5-day observation intervals.

### 3.2. Development Time

The development of *H. hampei* was completed at temperatures between 18 and 27 °C; its larvae could not survive and develop successfully at a temperature of 30 °C (Table 2). The longest development time in the egg stage was 10.16 days at 18 °C, while the shortest was 2.51 days at 30 °C. At 27 °C, the development times of larvae and pupae were the shortest, at 19.15 and 4.58 days, respectively. At 18 °C, the development times of larvae and pupae increased approximately 2-fold to 37.97 and 11.40 days, respectively, which were the longest. The range of the duration of the immature stage was 27.64–59.85 days at 27–18 °C, respectively. As demonstrated in Table 2, the temperature significantly affected the development time (*p* < 0.0001), which decreased with increasing temperature.

The development time of females in each stage was shortened significantly with increasing temperature (egg: *F*_3, 214_ = 180.46, *p* < 0.0001; lava: *F*_3, 214_ = 108.03, *p* < 0.0001; pupa: *F*_3, 214_ = 817.16, *p* < 0.0001; immature: *F*_3, 214_ = 252.04, *p* < 0.0001). The development time of males at 21, 24, and 27 °C was significantly shorter than that at 18 °C (egg: *F*_3, 38_ = 24.51, *p* < 0.0001; lava: *F*_3, 38_ = 8.49, *p* = 0.0002; pupa: *F*_3, 38_ = 276.34, *p* < 0.0001; immature: *F*_3, 38_ = 33.69, *p* < 0.0001); however, there were no significant differences among them. In addition, each stage of the females and males was compared at the same temperature (Table 3). The development times of the egg and pupal stages were similar between the females and males at each temperature. However, the development times of the larval stage were significantly different between the females and males at 18 and 21 °C (18 °C: *F*_1, 60_ = 26.89, *p* < 0.0001; 21 °C: *F*_1, 78_ = 17.34, *p* < 0.0001). The immature stage showed the same trend as the larval stage (18 °C: *F*_1, 60_ = 20.69, *p* < 0.0001; 21 °C: *F*_1, 78_ = 15.53, *p* = 0.0002). Furthermore, the difference in the development time between female and male larvae decreased with increasing temperature (Table 3).

### 3.3. Development Rate

Although the coefficient of determination for the linear regression equations effectively explains the relationship between temperature and the development rates of the egg and larval stages (egg: *R*^2^ = 0.8314; larva: *R*^2^ = 0.8887), no significant linear relationship was observed (egg: *p* = 0.0882; larva: *p* = 0.0573) (Table 4). However, a significant linear relationship was identified between temperature and the development rates of the pupa and immature stages (pupa: *p* = 0.0361; immature: *p* = 0.0352); the coefficients of determination for both were high (pupa: *R*^2^ = 0.9292; immature: *R*^2^ = 0.9309) (Table 4). Considering the effect of sex, the development rates of females and males were separated for linear regression analysis. For the female larvae, there was a significant linear relationship between temperature and the development rate, with a high coefficient of determination (*R*^2^ = 0.9569, *p* = 0.0218); in contrast, no significant linear relationship between temperature and the development rate of male larvae was observed (*R*^2^ = 0.4015, *p* = 0.3664) (Table 4). The lower developmental threshold (*T*_0_) and thermal summation (*K*) for the immature stage of *H. hampei* were 8.91 °C and 485.44 day-degrees (DD), respectively (Table 4).

### 3.4. Longevity and Fecundity

The sex ratios of *H. hampei* at 18–27 °C were between 4.21 and 6.27, with female adults being approximately five times as numerous as male adults. The temperature significantly affected female longevity (*F*_3, 214_ = 19.89, *p* < 0.0001), which was the longest at 18 °C and the shortest at 27 °C (Table 5). Although male longevity ranged widely from 12.50 to 26.50 days at temperatures of 18–27 °C, respectively, there were no significant differences between different temperatures (*F*_3, 38_ = 1.75, *p* = 0.1739) (Table 5). Regardless of the temperature, the longevities of female and male adults differed from one another significantly under the same conditions (18 °C: *F*_1, 60_ = 36.57, *p* < 0.0001; 21 °C: *F*_1, 78_ = 36.97, *p* < 0.0001; 24 °C: *F*_1, 71_ = 72.09, *p* < 0.0001; 27 °C: *F*_1, 43_ = 16.37, *p* = 0.0002) (Table 5).

At 18–27 °C, the adult pre-oviposition periods (APOPs) and total pre-oviposition periods (TPOPs) varied considerably with the temperature (Table 5). The number of oviposition days (Ovi-days) was the greatest at 24 °C, which was significantly different from that at other temperatures, except for 27 °C (Table 5). Notably, the number of oviposition days increased slightly with increasing temperature. The fecundities of the females were 8.18, 12.33, 29.00, and 21.84 eggs at temperatures of 18, 21, 24, and 27 °C, respectively, demonstrating significant differences among different temperatures (Table 5). The fecundity of *H. hampei* was the highest at 24 °C, while the fecundity at 18 °C was the lowest among all temperatures (Table 5).

### 3.5. Population Parameters

The age–stage-specific survival rate (*s_xj_*) of *H. hampei* was significantly affected by the temperature (Figure 2). Overall, the time span from the hatching of the first individual to the emergence of all individuals decreased with increasing temperature. At high temperatures, the adults tended to emerge relatively early. It took longer for all individuals to die at lower temperatures. At 18 °C, the population could survive for as long as 250 days before all individuals died, while this lasted a maximum of 150 days at 27 °C. Regardless of the temperature, the male adults emerged earlier than the female adults, and the female adults survived longer than the male adults (Figure 2). The age-specific survival rate (*l_x_*) decreased significantly with increasing temperature. The curve of the age-specific survival rate decreased sharply at high temperatures but decreased gradually at low temperatures (Figure 3).

The temperature also significantly influenced the initial reproduction time of the population. As demonstrated by the age-specific fecundity (*m_x_*), the higher temperature led to an earlier onset of reproduction (Figure 3). The first individual started ovipositing on day 63 at 18 °C, while it started on day 21 at 27 °C. The age–stage-specific fecundity (*f_x_*_4_) was higher during the early reproduction period of the entire population, regardless of the temperature (Figure 3). Overall, age-specific maternity (*l_x_m_x_*) was higher at high temperatures. Although peaks of age-specific fecundity (*m_x_*) and age–stage-specific fecundity (*f_x_*_4_) were observed during the late stage, the age-specific survival rate (*l_x_*) was very low in this stage. Therefore, the trend of age-specific maternity (*l_x_m_x_*) accurately indicated the potential and peak of reproduction for the population. As shown in the curve of age-specific maternity (*l_x_m_x_*), the reproduction period of *H. hampei* at 24 °C was rather long, and the fecundity remained stable at around 0.20 eggs (Figure 3). At 18 °C, the curve of age-specific maternity (*l_x_m_x_*) showed considerable changes with age and was generally lower than that at other temperatures; it also displayed two extreme reproduction peaks (Figure 3).

The net reproductive rate (*R*_0_), intrinsic rate of increase (*r*), finite rate of increase (*λ*) and mean generation time (*T*) were significantly different at different temperatures (Table 6). The net reproductive rate (*R*_0_) was the highest at 24 °C, reaching 13.32 eggs/individual. The intrinsic rate of increase (*r*) and finite rate of increase (*λ*) were the highest at 27 °C, with respective values of 0.0401 and 1.0409 day^−1^, whereas the parameters were the lowest at 18 °C, with respective values of 0.0036 and 1.0036 day^−1^. The rate of population growth increased with increasing temperature, and significant differences were observed between different temperatures except for between 21 and 24 °C (Table 6). The mean generation time (*T*) was the longest at 18 °C, and was 2.5 times as long as that at 27 °C. However, even at 27 °C, the mean generation time (*T*) still lasted about 50 days (Table 6).

### 3.6. Life Expectancy and Reproductive Value

The age–stage-specific life expectancy (*e_xj_*) of *H. hampei* at different temperatures is illustrated in Figure 4. At 18, 21, 24 and 27 °C, the life expectancy (*e_xj_*) of newly laid eggs was 118.48, 106.73, 87.89 and 45.89 days, respectively. However, the life expectancy (*e_xj_*) was only 8.73 days at 30 °C, which was significantly shorter than at other temperatures. Moreover, the life expectancy (*e_xj_*) at 21 °C was the longest only for newly laid eggs, while newly emerged female adults had the longest life expectancy (*e_xj_*) at other temperatures (Figure 4). Overall, higher temperatures led to a lower life expectancy. The life expectancy (*e_xj_*) of female adults was considerably longer than that of male adults at all temperatures; the difference between genders was especially pronounced at the beginning of the adult stage (Figure 4).

The peak of age–stage-specific reproductive value (*v_xj_*) was 2.2632, 4.4156, 8.7420 and 9.4402 at 18, 21, 24, and 27 °C, respectively. The peak value at 27 °C was 4 times as high as that at 18 °C, and the period of high reproduction was also longer at 27 °C than at 18 °C (Figure 5). The age–stage-specific reproductive value (*v_xj_*) increased significantly at the start of oviposition, regardless of the temperature (Figure 5). The age when the initial increase in the reproductive value occurred was earlier at higher temperatures; specifically, the increase started on the 52nd, 27th, 25th and 20th days at temperatures of 18, 21, 24, and 27 °C, respectively. Subsequently, the reproductive values peaked on the 67th, 36th, 98th, and 31st days at temperatures of 18, 21, 24, and 27 °C, respectively.

## 4. Discussion

The coffee berry borer has a special ecological habit. Most stages of the life cycle are completed inside coffee berries. Under the natural conditions of the field, they are practically unaffected by external forces because of the protection offered by the berries. However, many indoor experiments require day-by-day observation and recording of its behaviors, inevitably disturbing the lives of *H. hampei* and affecting the experimental results. In this study, we compared different observation intervals to understand the effect of these conditions on the survival and reproduction of *H. hampei* in the laboratory. We found that its survival and reproduction were high when the observation interval was long. An observation interval of 10 days increased reproduction by as much as 35 times compared to daily observation. Accordingly, disturbance by observation certainly affected the survival and reproduction of this pest. With the OB1 treatment, female adults hardly burrowed into the artificial diet, and only stayed on the surface to eat. At the same time, frequent replacement of the artificial diet caused the tunnels in the diet to be relatively shallow. Therefore, frequent observations and disturbances caused by human manipulation when rearing *H. hampei* in the laboratory do indeed affect the performance of survival and reproduction. We discussed the effect of the observation intervals on survival and reproduction and clarified the conditions for stable rearing, enabling the results to more accurately reflect the actual fecundity.

Difficulties were encountered in the indoor experiment on *H. hampei* because of its unique ecological habits. Our novel study verified that the length of the observation interval significantly affected survival and reproduction. Although a longer observation interval led to an improved survival rate and fecundity, the length of the observation interval also affected the information on the daily change in individuals. As demonstrated by the survival rate curves, longer intervals allowed more satisfactory survival, but prevented effective evaluation of the daily changes in the survival rate (Figure 1), and the fecundity was the same. Observation intervals that were too long prevented and accurate analysis of the survival rate and fecundity at the time scale “age”. Therefore, the observation interval was set as 5 days for the longevity and fecundity of adults in the follow-up life history experiments. This provided improved and stable survival rate and fecundity data compared to those acquired using a 1-day observation interval.

The ecological habits of *H. hampei* regarding damage inside coffee berries make life history experiments difficult and challenging [31]. Therefore, studies on the basic ecology of this pest remain scant. In the past, the development of the immature stage was observed by regularly sampling and dissecting infested fruit in the field in Mexico [32] and Colombia [13], and in coffee farms across an elevational gradient in Hawaii [15]. Later, artificial diets were developed for the mass rearing of *H. hampei* [12,16,21]. This made the observation process of its life history simpler and easier. In the laboratory, artificial infection of coffee berries also facilitated daily observation of the development of coffee berry borer [33]. Azrag et al. (2020) dug tiny holes in fresh coffee berries, inserted larvae, and covered the seeds with tinfoil to simulate the dark and humid environments within the berries for life history observation [31]. This approach differed from observation by dissection. It not only allowed an accurate understanding of the development of the individual through single rearing, but also made it convenient to perform the observation of numerous individual samples. These studies employed coffee berries as the food source to rear *H. hampei* [31,33]. However, several uncertain factors affect the use of these berries in experiments, such as varying degrees of maturity and changes in berry quality over time. Moreover, coffee berries are not harvested all year round. Therefore, coffee berries cannot be stably supplied for conducting numerous studies on long-term life history. An artificial diet provides a convenient and stable food source and simplifies the rearing process. Therefore, our study applied the artificial diet reported by Brun et al. (1993) [21] and reared the pests individually to observe their life history, enabling an accurate understanding of the development of each individual and simplifying the process of observation. Moreover, using the artificial diet can prevent damage or disturbance caused by dissecting the berries.

Jaramillo et al. (2009) [33] and Azrag et al. (2020) [31] conducted a temperature-dependent life history experiment with coffee berries. In these two studies, the coffee berry borers completed their life cycles in temperature ranges of 20–30 °C and 18–30 °C, respectively [31,33]. At 30 °C, the development times of the immature stage were 23.3 and 18.00 days. However, the borers could not successfully survive at 30 °C in our study. Further observation revealed that the artificial diet deteriorated at high temperatures, meaning the failure in survival was probably caused by the food source. In addition, larvae eating inside berries have satisfactory shelters, and the temperature inside the berries also differs slightly from that of the external environment. On the other hand, larvae feeding on an artificial diet can only eat on the surface and lack protection from high-temperature environments. Nevertheless, whether this is the reason for the failure to successfully survive remains to be confirmed by related studies.

At temperatures of 18–30 °C, the development times of eggs were between 2.51 and 10.46 days, which were shortened with increasing temperature. Brun et al. (1993) reported that the egg period when reared on an artificial diet was about 5 days at a constant temperature of 25 °C [21]. In our study, the development times of eggs (5.65 days) at 24 °C were close to this value. The other report used coffee berries as the food source, revealing that the egg period lasted 4.62–11.58 days at temperatures in the range of 18–30 °C [31]. However, our results for the egg period were shorter than those in the same temperature range. This minor variation may be caused by the method of egg collection and the conditions in the incubator. The larval development time in our study was 37.97 days at 18 °C, which was similar to the reported duration of 39.45 days [31]. At 27 °C, two other reports have indicated development times of larva of 12.0 and 13.81 days [33,34]. The larval period (19.15 days) reported in this study was significantly longer than those. The variations among these results may be due to the different food sources, as well as rearing methods and conditions. The overall results reported in this study are consistent with the concept of thermal summation in insects, that is, higher temperatures lead to shorter development times. In our study, the development times for pupae were 4.58–11.40 days at temperatures of 18–30 °C, consistent with the findings by Azrag et al. (2020) [31]. The lengths of the pupal stage at 24 and 27 °C were similar to the pupal development time reported in most studies [21,34,35,36]. On the other hand, the pupal stage was 16.3 days at 20 °C when coffee berries were used for rearing [33]. However, the pupal stages we observed at 18 and 21 °C were 11.40 and 6.20 days, respectively, which were significantly shorter than the above result. Insects are not capable of eating and other activities during their pupal stages. The research of Hamilton et al. (2019) pointed out that the development time from egg to adult is 46.7–49.5 days at 20 °C, and 32.1–36.9 days at 23 °C [15]. The trends in this study when comparing with the development time are the same. They only consume and convert the nutrition they acquire as larvae for metamorphosis. Theoretically, similar temperature conditions should be close in terms of their effects on pupal stage development rates. Therefore, the great variation between the results reported for the development times of pupae may be attributable to differences in food sources and nutrient composition or consumption during the larval stage.

The development rate at various temperatures can be obtained via life history experiments, and biologically relevant information such as the lower developmental threshold and thermal summation can be analyzed theoretically, so as to further understand and estimate the effect of temperature on the development, distribution, and population dynamics of pests [37,38]. In one report, the relationship between the development rate and temperature in *H. hampei* was analyzed using linear regression [33]. Another report used ILCYM software [39] for analysis to select the most suitable nonlinear model and describe the development rate and mortality of *H. hampei* at each instar [31]. Many studies on insect species have used linear regression to analyze the relationship between temperature and the development rate. This approach has often been employed to estimate the lower developmental threshold and thermal summation [40,41,42]. With the use of coffee berries as the food source for rearing *H. hampei*, the lower developmental thresholds of the immature stage were 14.9, 13.03 and 13.9 °C, and the thermal summations were 262.47, 312.50 and 386 degree-days, respectively [15,31,33]. In addition, in a field investigation, under the lower developmental threshold of 16.5 °C, the thermal summation also required 225 degree-days [43]. In this study, we used an artificial diet as the food source, and the lower developmental threshold and thermal summation of the immature stage were 8.91 °C and 485.44 degree-days, respectively. Our results differ substantially from the other reports, possibly due to differences in food sources. Furthermore, the *H. hampei* colonies used in the aforementioned studies differed in their regions of origin, which varied considerably in their climate conditions. According to the monthly average temperature of the coffee production area in Taiwan in 2020, the generation number of *H. hampei* could reach in about 11 generations per year. In Hawaii, this value is 2.11–3.27 generations/season at high altitudes, and 4.13–4.96 generations/season at low altitudes [15]. One report indicated that *H. hampei* has eight to nine generations per year in Uganda, and only two or three generations in Colombia [4]. In Brazil, the coffee berry borer can produce 5.09–10.53 generations annually [44]. Consequently, the number of generations occurring in Taiwan per year is closer to and slightly higher than that in Uganda and Brazil.

This study used linear regression on the development rate of larvae, whose coefficient of determination exhibited considerable explanatory power (*R*^2^ = 0.8887). We attempted separate linear regression analyses for female and male larvae. The results of the linear regression analysis showed that the influence of males was significant for the overall larvae (including females and males). This not only deviated from the linear relationship, but also led to an underestimation of the lower developmental threshold (*T*_0_ = 7.31) and an overestimation of the thermal summation (*K* = 357.14) for the overall larvae. The linear regression equation of the female larvae was closer to the actual development rate of larvae. The lower developmental threshold and thermal summation of female larvae were estimated as 9.56 °C and 308.64 degree-days, respectively. These results should be a more precise estimate than an analysis with overall larvae. Hamilton et al. (2019) used the lower developmental threshold of a research report to estimate the thermal summation in the field, which required 332 and 386 degree [15]. In fact, our experiments, including the linear analysis, mainly only used the data of foue observations, and the temperature range was limited (18–27 °C). The growth and development of insects occur in environments with fluctuating temperatures. Obtaining life history information at a constant temperature will result in slight deviations. Milosavljevic et al. (2020) used six constant temperature and variable temperature conditions for the growth and development of *Diaphorina citri*, and the development period was significantly longer under the variable temperature environment than at constant temperature [45]. It is true that the variable temperature method can better meet the growth and development conditions of insects in nature, and can estimate a more accurate developmental period. The development time obtained inour CBB constant-temperature experiment should also be shorter than that in the actual natural environment with variable temperature; this is where the results should still be paid attention to in inference and application.

Few comprehensive studies have been published worldwide on the longevity and fecundity of *H. hampei* [12,43]. This pest can live for over 3 months, and some have lived for up to 8 months [43]. It has also been reported that the longevity of males is about 20–87 days, and that of females could be as long as 157 days [4]. The longevity we obtained for females and males was, respectively, 53.42–115.77 and 14.29–26.50 days in the temperature range of 18–27 °C. These results are shorter than those reported in the mentioned studies. We also observed a significant difference in longevity between females and males. The longevity of females was substantially longer than that of males. The females could survive for 3–4 months at 24 °C, and even under a high temperature of 27 °C, they could still survive for about 2 months. In this study, the number of oviposition days was 32.67 and the fecundity was 29.00 eggs/female at 24 °C, demonstrating better reproduction performance. Conversely, the number of oviposition days was 18.82 and the fecundity was only 8.18 eggs/female at the low temperature of 18 °C. A report indicated that the oviposition period is 40 days and the fecundity is 56 eggs/female, which are values significantly higher than the results obtained in this study [46]. Another report found that the oviposition period and the fecundity were 11–15 days and 24–63 eggs/female, respectively, and the maximum oviposition of individuals could reach 119 eggs [47]. There are also research results that indicate that *H. hampei* can have a fecundity of 35.6 eggs/female in the field environment [12]. Our results for fecundity were closer to these, but the maximum oviposition of the individuals was only 91 eggs, and the number of oviposition days was comparatively higher.

The two-sex life table theory based on age–stage can accurately interpret the development, survival, reproduction, and population performances of insects [18,19]. Unlike the female life table, the two-sex life table considers the contributions of all the individuals (immature stage, female adults and male adults) to the population [20]. It also clearly describes the stages of insects and their age–stage-specific survival performance [17,48]. In this study, notable overlaps of the age–stage-specific survival rate (*s_xj_*) curves in different stages were noted (Figure 2), indicating significant differences in the development rates of individuals within a population. The range of age with high overlaps was shortened with increasing temperature. The beginning overlapping age was earlier at high temperatures than at low temperatures, indicating a negative correlation between the development time and temperature. Moreover, it revealed a high stage and generation overlap in the population of *H. hampei*. Overall, its survival rate decreased gradually with age, and a higher temperature led to a sharper drop in the survival rate. When it emerged as an adult, its age–stage-specific survival rate (*s_xj_*) rose and then gradually slowed down with an increase (Figure 2). This showed that the survival rate was considerably stable at the beginning of the adult stage, and the mortality rate was generally low after emergence. At 27 °C, about 40% of *H. hampei* emerged successfully, which was the lowest percentage among all the test temperatures. Although the development time of the immature stage lasted for up to 2 months at the low temperature of 18 °C, its survival was relatively stable. According to the age–stage-specific survival rate (*s_xj_*), the suitable temperature for the development of this pest was 21 °C, at which about 85% of the individuals successfully emerged as adults.

The age-specific survival rate (*l_x_*) differed significantly with temperature and corresponded to the age-specific fecundity (*m_x_*) (Figure 3). Except for 24 °C, this pest displayed high fecundity at the beginning of the adult stage, during which its survival rate was also relatively stable. Moreover, although the age-specific fecundity (*m_x_*) showed a peak at the end of adulthood, it did not indicate that the peak of reproduction occurred during the final period of the population’s development. Overall, the population of *H. hampei* could reproduce stably and over the long term at 24 °C. However, at low or high temperatures, they would oviposit intermittently and aperiodically, and the fecundity of each individual would not decrease near the end of its life. This reproductive characteristic may be a critical piece of information for pest management. This indirectly indicated that female adults were the key sex affecting the increase in population in pest management. Effectively decreasing the density of female adults might profoundly reduce the fecundity of the entire population.

We examined the temperature-dependent population performances of *H. hampei*. The net reproductive rate (*R*_0_) was the highest at 24 °C (13.32 eggs/individual). The intrinsic (*r*) and finite (*λ*) rates of increase were the highest at 27 °C with respective values of 0.0401 and 1.0409 day^−1^. This means that when the time tended to infinity and the population reached a stable age–stage distribution (SASD), the population increased 1.0409 times per day. When the initial population was 10 and the time was infinite, the population increased by approximately 110 times by the 60th day. The population increased about 11 times in two months. The mean generation time (*T*) was the shortest at 27 °C (51.34 days). Provided that the population increased stably, it required approximately 50 days to reach 7.83 times (*R*_0_ times) at 27 °C. There were some reports that preliminarily estimated the population parameters of this pest based on the results of field investigations. They reported the net reproductive rate (*R*_0_), intrinsic rate of increase (*r*), finite rate of increase (*λ*) and mean generation time (*T*) to be 25.0 eggs/individual, 0.065 day^−1^, 1.067 day^−1^, and 45.2 days, respectively [32]. In Colombia, continuous feeding with an artificial diets, the net reproductive rate (*R*_0_), intrinsic rate of increase (*r*), finite rate of increase (*λ*), and mean generation time (*T*) were 19.3 eggs/individual, 0.068 day^−1^, 1.070 day^−1^, and 45.4 days, respectively [12]. These parameters differed slightly from the results of our study. Furthermore, many biological and nonbiological factors in the field cnaaffect the performances of populations. Therefore, more comprehensive experiments arerequired to understand the population information of this pest.

Understanding this characteristic of populations of *H. hampei* is crucial for pest management. In particular, it is especially important to note the control measures for female adults. The age–stage-specific reproductive value (*v_xj_*) indicates the expected contribution of an individual to its population at a specific age (*x*) and stage (*j*) [26,27]. The reproductive value (*v_xj_*) of a newborn egg (*v*_01_) represents the finite rate of increase (*λ*) for the population. In this study, the peak of the reproductive value occurred on the 67th, 36th, 98th, and 31st days at 18, 21, 24, and 27 °C, respectively (Figure 5). Accordingly, the female adults at these ages exhibited a higher expected contribution to the population than individuals in the other stages or female adults of other ages. As demonstrated in the reproductive value (*v_xj_*), this pest exhibited high reproductive potentials at 24 and 27 °C. Accordingly, although the reproduction ability of *H. hampei* was not particularly strong at low temperature (18 °C), it was capable of continual reproduction for nearly half a year. This is important information for its population characteristics.

In summary, the reproduction of *H. hampei* was not particularly high, and the population did not increase rapidly. However, the period of the larva and adult stages with damaging ability constituted 90% of its lifespan. This means that most individuals of the field populations were in stages that could directly damage coffee berries. On the other hand, as 80% of the adults were female, the reproductive potential was astonishing. Moreover, female adults could live for more than 2 months, and the reproduction period of the entire population could last for as long as half a year. If this pest successfully establishes a population in a coffee farm, it would incur lasting damage to the crops, leading to severe economic loss. This study discussed the biology of *H. hampei*, including the effect of different observation intervals on their survival and reproduction, as well as their temperature-dependent life history and population performances, thereby providing a comprehensive study on the biological and population performances of this pest.

This research report completed the life history at a constant temperature in Taiwan and obtained the population parameters. Although Taiwan is not a major coffee production area in the world, it is located at the junction of tropical and subtropical regions. The research results can provide important reference information for IPM of *H. hampei*. In addition, they can also be used for the growth of populations and the forecasting of the occurrence of populations as important basic information for the control of this pest.

## Figures and Tables

**Figure 1 insects-14-00499-f001:**
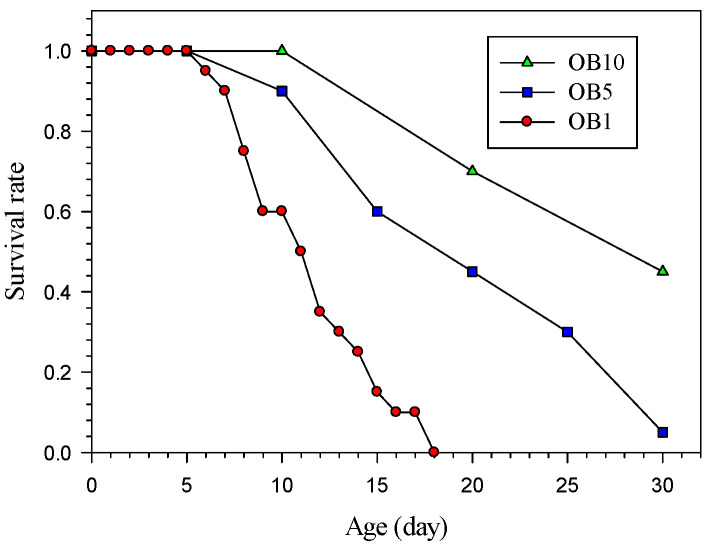
Survival rate of *Hypothenemus hampei* (females) at different observation intervals.

**Figure 2 insects-14-00499-f002:**
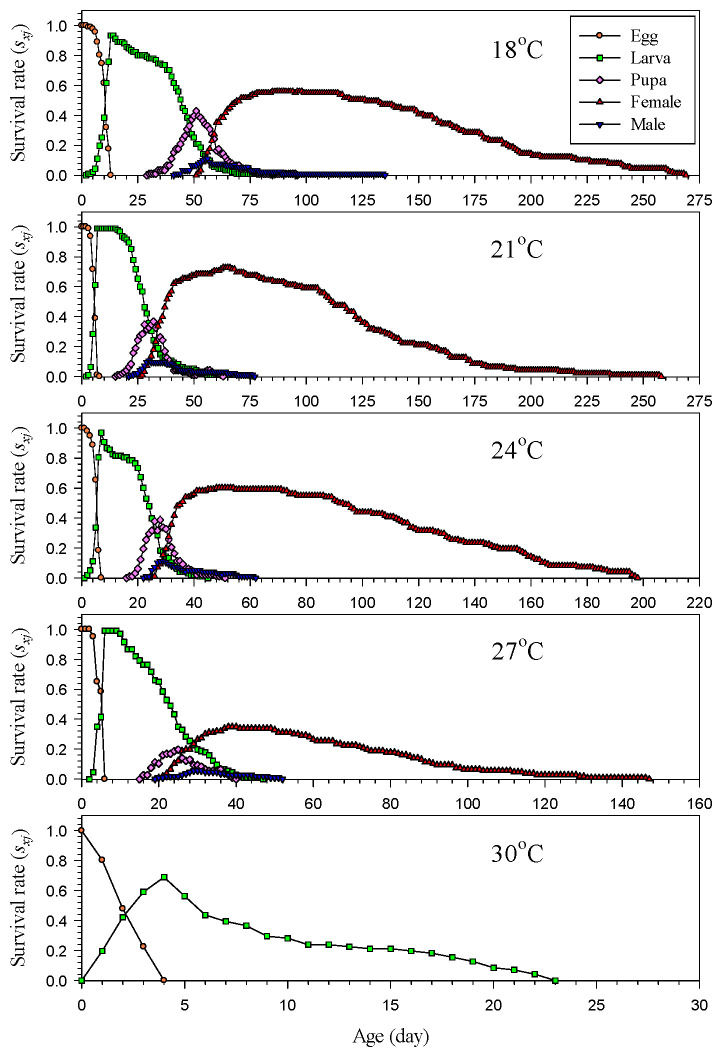
Age–stage-specific survival rate (*s_xj_*) of *Hypothenemus hampei* reared on an artificial diet at various temperatures.

**Figure 3 insects-14-00499-f003:**
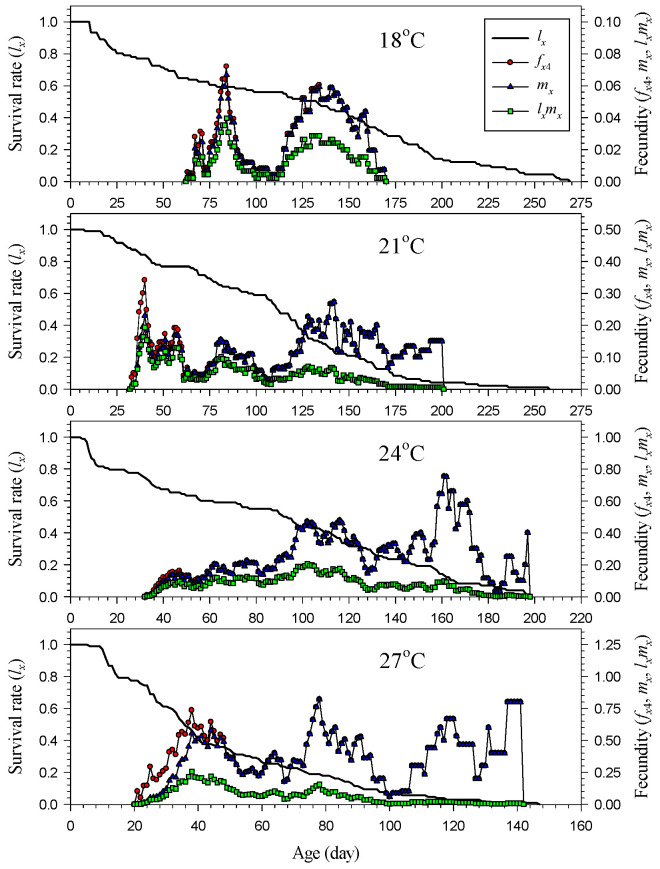
Age-specific survival rate (*l_x_*), fecundity (*m_x_*), maternity (*l_x_m_x_*) and age–stage-specific fecundity (*f_x_*_4_) of *Hypothenemus hampei* reared on an artificial diet at various temperatures.

**Figure 4 insects-14-00499-f004:**
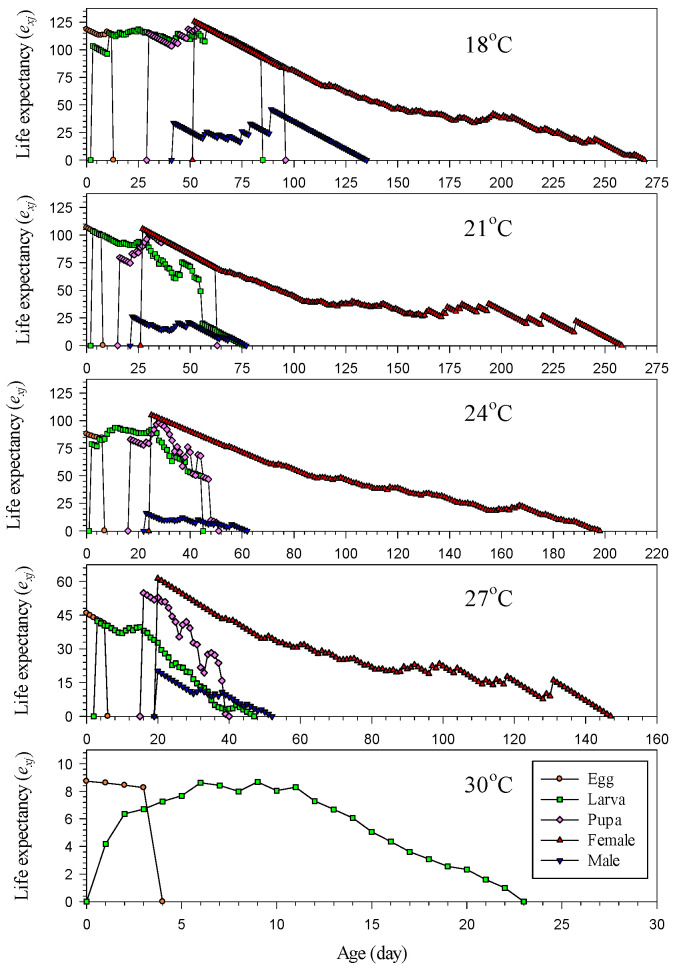
Age–stage-specific life expectancy (*e_xj_*) of *Hypothenemus hampei* reared on an artificial diet at various temperatures.

**Figure 5 insects-14-00499-f005:**
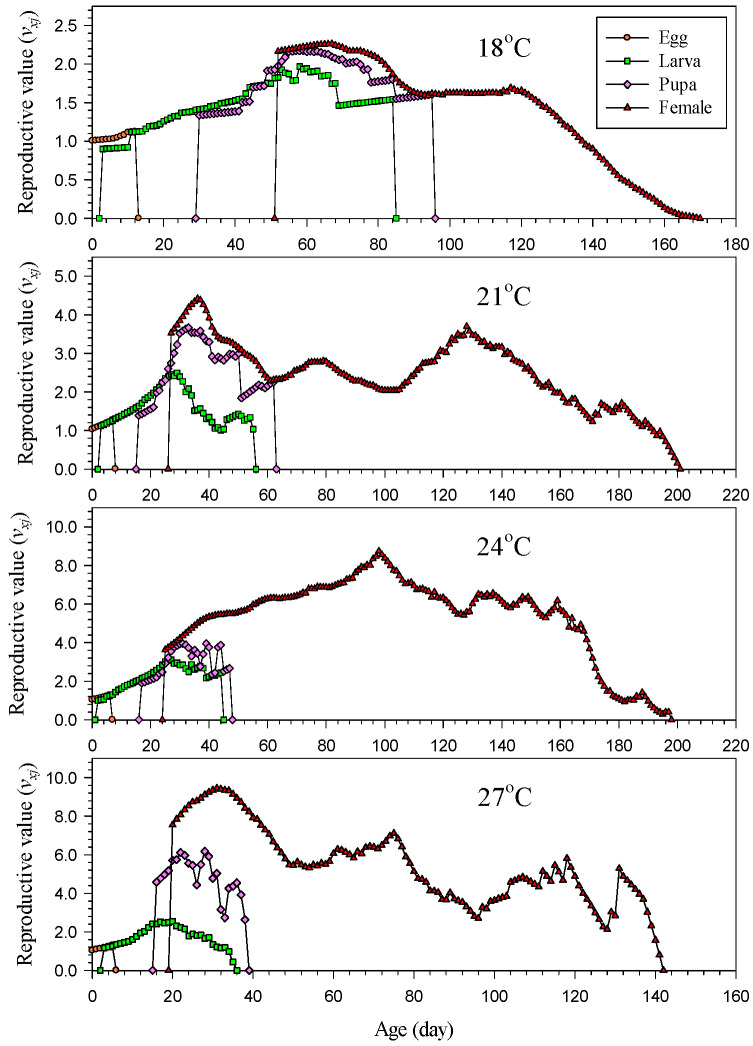
Age–stage-specific reproductive value (*v_xj_*) of *Hypothenemus hampei* reared on an artificial diet at various temperatures.

**Table 1 insects-14-00499-t001:** Cumulative fecundity of *Hypothenemus hampei* at different observation intervals during a 30-day period.

Interval for Observation	Cumulative Fecundity (Eggs/Female)		
	Mean ± SE		
	0–10 Days	0–20 Days	0–30 Days
OB1 ^1^	0.40 ± 0.26 c ^2^	0.55 ± 0.29 c	0.55 ± 0.29 c
OB5	5.95 ± 1.39 b	7.95 ± 1.83 b	8.45 ± 1.96 b
OB10	12.95 ± 1.45 a	14.60 ± 1.64 a	19.35 ± 2.57 a
*F*	28.82	24.18	25.37
*df*	2, 57	2, 57	2, 57
*P*	<0.0001	<0.0001	<0.0001

^1^ OB1, OB5, and OB10 had 1-, 5-, and 10-day intervals for observation during the experiment period, respectively. ^2^ Means within each column followed by different letters are significantly different (Tukey’s HSD, *p* < 0.05).

**Table 2 insects-14-00499-t002:** Developmental duration of *Hypothenemus hampei* reared on an artificial diet at various temperatures.

Temperature (°C)	Developmental Duration (Day)							
	Mean ± SE							
	*n*	Egg	*n*	Larva	*n*	Pupa	*n*	Immature
18	91	10.46 ± 0.22 d	62	37.97 ± 1.17 c	62	11.40 ± 0.14 d	62	59.85 ± 1.17 d
21	95	6.04 ± 0.10 c	81	23.48 ± 0.88 b	80	6.20 ± 0.06 c	80	35.86 ± 0.89 c
24	98	5.65 ± 0.11b c	75	20.45 ± 0.62 ab	73	5.71 ± 0.05 b	73	31.93 ± 0.63 b
27	106	5.19 ± 0.10 b	54	19.15 ± 0.75 a	45	4.58 ± 0.11 a	45	27.64 ± 0.71 a
30	71	2.51 ± 0.13 a		-		-		-
*F*		397.32		89.57		1045.92		239.19
*df*		4, 456		3, 268		3, 256		3, 256
*P*		<0.0001		<0.0001		<0.0001		<0.0001

Means within each column followed by different letters are significantly different (Tukey’s HSD, *p* < 0.05).

**Table 3 insects-14-00499-t003:** Developmental duration of female and male *Hypothenemus hampei* reared on an artificial diet at various temperatures.

Temperature (°C)	Sex	Developmental Duration (Day)				
		Mean ± SE				
		*n*	Egg	Larva	Pupa	Immature
18	♀♀	52	10.29 ± 0.25 a	40.19 ± 1.15 b	11.40 ± 0.16 a	61.88 ± 1.18 b
	♂♂	10	11.50 ± 1.07 a	26.40 ± 0.86 a	11.40 ± 0.22 a	49.30 ± 1.36 a
21	♀♀	69	6.00 ± 0.12 a	24.91 ± 0.92 b	6.23 ± 0.07 a	37.14 ± 0.92 b
	♂♂	11	6.73 ± 0.27 b	15.09 ± 1.10 a	6.00 ± 0.13 a	27.82 ± 1.16 a
24	♀♀	59	5.95 ± 0.13 a	20.73 ± 0.57 a	5.78 ± 0.05 b	32.46 ± 0.58 a
	♂♂	14	5.86 ± 0.14 a	18.43 ± 2.21 a	5.43 ± 0.14 a	29.71 ± 2.20 a
27	♀♀	38	4.97 ± 0.18 a	18.32 ± 0.82 a	4.63 ± 0.12 a	27.92 ± 0.80 a
	♂♂	7	5.86 ± 0.14 b	16.00 ± 1.40 a	4.29 ± 0.29 a	26.14 ± 1.47 a

Means of females and males in each stage at the same temperature followed by different letters are significantly different (Tukey’s HSD, *p* < 0.05).

**Table 4 insects-14-00499-t004:** Estimates of linear regression analysis describing the effect of temperature on the development rate, lower developmental threshold and thermal summation of *Hypothenemus hampei*.

Sex	Stage	Regression Equation	*p*	*R* ^2^	*T*_0_ (°C)	*K* (DD)
♀♀ + ♂♂	Egg	$r\left(T\right)=-0.06943+0.01009T$	0.0882	0.8314	6.88	99.11
	Larva	$r\left(T\right)=-0.02046+0.00280T$	0.0573	0.8887	7.31	357.14
	Pupa	$r\left(T\right)=-0.14390+0.01353T$	0.0361	0.9292	10.64	73.91
	Immature	$r\left(T\right)=-0.01835+0.00206T$	0.0352	0.9309	8.91	485.44
♀♀	Egg	$r\left(T\right)=-0.07650+0.01043T$	0.0752	0.8553	5.65	73.91
	Larva	$r\left(T\right)=-0.03098+0.00324T$	0.0218	0.9569	9.56	308.64
	Pupa	$r\left(T\right)=-0.13864+0.01324T$	0.0379	0.9257	10.47	75.53
	Immature	$r\left(T\right)=-0.01972+0.00210T$	0.0278	0.9451	9.39	476.19
♂♂	Egg	$r\left(T\right)=-0.06079+0.00911T$	0.1087	0.7945	6.67	109.77
	Larva	$r\left(T\right)=0.00883+0.00206T$	0.3664	0.4015	−4.29	485.44
	Pupa	$r\left(T\right)=-0.17281+0.01515T$	0.0303	0.9404	11.41	66.01
	Immature	$r\left(T\right)=-0.00667+0.00172T$	0.1729	0.6841	3.88	581.40

$r\left(T\right)=a+bT$, where $r\left(T\right)$ is the development rate (day^−1^), and *T* is the temperature (°C). *T*_0_, lower developmental threshold (°C); *K*, thermal summation (DD).

**Table 5 insects-14-00499-t005:** Longevity and fecundity of *Hypothenemus hampei* reared on an artificial diet at various temperatures.

Temperature (°C)	Longevity (Day)				Parameters of Fecundity				
	Mean ± SE				Mean ± SE				
	*n*	♀♀	*n*	♂♂	*n*	APOP ^1^ (Day)	TPOP ^2^ (Day)	Ovi-Days ^3^ (Day)	Fecundity (Eggs/Female)
18	52	115.77 ± 6.26 aA	10	26.50 ± 7.89 aB	17	23.82 ± 4.80 b	83.06 ± 4.51 c	18.82 ± 4.67 b	8.18 ± 2.70 c
21	69	95.51 ± 4.83 bA	11	20.91 ± 4.41 aB	61	14.75 ± 2.49 ab	51.54 ± 2.63 b	19.75 ± 2.39 b	12.33 ± 2.03 bc
24	59	97.63 ± 4.81 abA	14	12.50 ± 2.81 aB	45	23.00 ± 3.48 b	56.29 ± 3.41 b	32.67 ± 3.36 a	29.00 ± 3.50 a
27	38	53.42 ± 4.07 cA	7	14.29 ± 3.35 aB	38	8.95 ± 1.91 a	36.87 ± 2.17 a	21.71 ± 2.31 ab	21.84 ± 2.67 ab
*F*		19.89		1.75		4.65	22.36	4.74	9.70
*df*		3, 214		3, 38		3, 157	3, 157	3, 157	3, 157
*P*		<0.0001		0.1739		0.0038	<0.0001	0.0034	<0.0001

Means within each column followed by different lowercase letters, and longevity values of females and males at the same temperature followed by different capital letters are significantly different (Tukey’s HSD, *p* < 0.05). ^1^ APOP: adult pre-oviposition period of female adults. ^2^ TPOP: total pre-oviposition period of females counted from birth. ^3^ Ovi-days: oviposition days.

**Table 6 insects-14-00499-t006:** Population parameters of *Hypothenemus hampei* reared on an artificial diet at various temperatures.

Temperature (°C)	Population Parameters			
	Mean ± SE			
	Net Reproductive Rate *R*_0_ (Eggs/Individual)	Intrinsic Rate of Increase *r* (Day ^−1^)	Finite Rate of Increase *λ* (Day ^−1^)	Mean Generation Time *T* (Day)
18	1.53 ± 0.59 c	0.0036 ± 0.0037 c	1.0036 ± 0.0037 c	117.57 ± 13.08 d
21	7.92 ± 1.43 b	0.0284 ± 0.0025 b	1.0288 ± 0.0026 b	72.89 ± 4.68 b
24	13.32 ± 2.15 a	0.0297 ± 0.0021 b	1.0301 ± 0.0022 b	87.28 ± 4.36 c
27	7.83 ± 1.39 b	0.0401 ± 0.0040 a	1.0409 ± 0.0041 a	51.34 ± 2.53 a

Standard errors were estimated using 100,000 bootstrap resampling. Means within each column followed by different letters are significantly different according to a paired bootstrap test (*B* = 100,000 resampling).

## Data Availability

Publicly available datasets were analyzed in this study. This data can be found here: https://reurl.cc/QX3K5o/ (accessed on 24 May 2023).

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
