# Peer review of "Temperature-Dependent Biology and Population Performances of the Coffee Berry Borer Hypothenemus hampei (Ferrari) (Coleoptera: Curculionidae: Scolytinae) on Artificial Diet"

_insects, 2023, doi:10.3390/insects14060499_

Round 1
Reviewer 1 Report
Overall, this is an interesting manuscript. It addresses several aspects related to CBB life history including survival, reproduction, pre-oviposition, developmental stages, longevity etc. The methodology used is appropriated, results are clearly presented. The discussion is good, but it can be improved. The following are the detailed recommendations.
Comments, questions, suggestions all of them are intended to improve the quality of this manuscript.
Line by line:
Title:.. add “on artificial diet.”
Line 11. … but also, the coffee beans in storage. This can happen, but only when the stored green coffee has not the correct moistures (10-12%) for storage. Moistures lower than 17% do not allow CBB reproduction. If you have the reference to support your statement, pleases include it or just delete it.
Keywords: Add: Survival rate, fecundity, longevity. Delete: coffee berry borer, it was included in the title, so it does not need to be repeat.
Line 50. Scientific names: include (Order and Family) for each of them.
Line 61-62. Those statements are not necessarily true. Yes, there are few references, but you must to include those reference here. Look that nobody has conducted any biology or life history of CBB around of the world.
Line 63-73. All of this is relevant, but you only cite 3 references about the life history experiments and the methodology proposed “age-stage, two -sex life table”. However, here you must include some reference about CBB life history parameters previously conducted under field conditions or laboratory.
Here you can see some references addressing this topic:
Ruiz C. and Baker S. P. (2010) Life table of Hypothenemus hampei (Ferrari) in relation to coffee berry phenology under Colombian field conditions. Scientia Agricola (Piracicaba, Braz.) 67, 658–668.
Hamilton, L. J., Hollingsworth, R. G., Sabado-Halpern, M., Manoukis, N.C., Follett, P. A., and Johnson,
M. A. (2019). Coffee berry borer (Hypothenemus hampei) (Coleoptera: Curculionidade) development across an elevational gradient on Hawaii Island: applying laboratory degree-day predictions to natural field populations. PLoS ONE 14:e0218321. doi: 10.1371/journal.pone.0218321
Vega F. E., Kramer M., Jaramillo J. (2011). Increasing coffee berry borer (Coleoptera: Scolytinae) female density in artificial diet decreases fecundity. J. Econ. Entomol. 104(1): 87-93.
Portilla M, Mumford J, Baker P. 2000. Reproductive potential response of continuous rearing of Hypothenemus hampei (Coleoptera: Scolytidae) on Cenibroca artificial diet. Revista Colombiana de Entomologia 26: 99–105.
Baker, P.S. The Coffee Berry Borer in Colombia: Final Report of the DFID-Cenicafé-CABI Bioscience IPM for Coffee Project (CNTR 795 93/1536A); CABI: Wallingford, Oxfordshire, UK, 1999; 144 pp.
You already have the last reference (Baker 1999), but please discuses in deep several aspects related to CBB life history and reproduction parameters that are included in this report.
Line 87. The artificial diet … Brun et al. (1993). Why this old diet was used, knowing the novel more effective and improved diets are used to rearing CBB and its parasitoids. Reports from 2000 until 2022.
In the following references you can see specific aspects about CBB reproduction on artificial diets, but also the potential use of those diets to rearing CBB parasitoids.
Portilla M, Mumford J, Baker P. 2000. Reproductive potential response of continuous rearing of Hypothenemus hampei (Coleoptera: Scolytidae) on Cenibroca artificial diet. Revista Colombiana de Entomologia 26: 99–105.
Portilla, M. and D. Streett. 2006. Nuevas t.cnicas de producci.n masiva de Hypothenemus hampei sobre la dieta artificial Cenibroca modificada. Rev. Col. Entomol. 57: 37–50 (Spanish)
Portilla M. & Grodowitz M. (2018). Abridged Life Tables for Cephalonomia stephanoderis and Prorops nasuta (Hymenoptera: Bethylidae) Parasitoids of Hypothenemus hampei (Coleoptera: Curculionidae: Scolytinae) Reared on Artificial Diet. Journal of Insect Science 18 (2): 20; 1-7.
Portilla M, Mumford J, Baker P. 2000. Reproductive potential response of continuous rearing of Hypothenemus hampei (Coleoptera: Scolytidae) on Cenibroca artificial diet. Revista Colombiana de Entomologia 26: 99–105.
Portilla M., Douglas S., (2022). Biological Responses of Hypothenemus hampei (Coleoptera Curculionidae) on Cenibroca Artificial Diet at Different Moisture Content Levels and Relative Humidities. Florida Entomologist, 105 (2): 137-144.
Line 92. This first laboratory trial was a good experiment to determine the minimal impact of CBB survival caused by the methodology (observation intervals) during evaluations.
Figure 1. Be specific: Hypothenemus hampei (Female). ?
Line 247. Correction. 27.64 – 59.85 days at 27 oC – 18 OC respectively.
Figure 2. This figure 2 is not needed. All regressions are included on table 4. It is no necessary to repeat data or information.
Line 373 and table 7. The mean generation time (T) for all different temperatures (18, 21, 24, and 27 oC) looks too long for me. How do you explain that? Those data suggest that a constant temperature between 3 and 7 generation of CBB per year under laboratory conditions. Correct?
From my experience rearing CBB and parasitoids we get about 1 generation of CBB each (38-45 days) at 24 oC
Line 531. Reference?
In discussion: There are several factors that may affect the CBB reproduction under field conditions and laboratory conditions different of temperature. For example, the humidity is relevant for fecundity and developmental stages. Under field conductions and when berries are used in the lab for rearing CBB, the moisture of berry is critical to allow CBB reproduction. Finally, the use of diets is important to allow a continuous rearing of CBB in the lab a long of the year, which is needed for multiple uses. For example, for allow rearing of CBB parasitoids or conduct additional laboratory experiments for supply CBB adults for field trials as well.
The age of developing berries is critical for CBB reproduction and the pre-oviposition time depends of it under field conditions. I understand that this work was conducted under laboratory conditions and it bring relevant information about all CBB life history parameters. However, in the discussion is important to include aspects about field conductions. For example, the number of CBB generations will dependent of many factors no only temperature. For example, flowering patter in some locations are different, which affect the distribution of berries along of the coffee season, and it affect the CBB reproduction and population as well. There is not the same planting coffee in Brazil, Colombia, Mexico, Hawaii, Tanzania, Kenya, India or Vietnam. Each location has its own weather conditions, which impact the coffee plant phenology and subsequently the CBB survival, reproduction ans population. So, those aspects need to be included in the discussion to see the implications of all your results not just for laboratory conductions and the potential of rearing CBB, but also, for rearing parasitoids or for facilitated better understanding CBB biology and its potential control (integrated pest management).
This is a good manuscript. It just needs to include those aspects previously disused. Good job!
Author Response
Reply to Reviwer1
Thank you very much for your professional(on CBB), careful and kind review of our manuscript, especially for providing us useful information and literature on CBB. This is really helpful making our manuscript better and more correct. We tried to revise our manuscript based on your comments point by point.
- Title:.. add “on artificial diet.”
Reply: Thank you for your comment. We have revised it.
- Line 11. … but also, the coffee beans in storage. This can happen, but only when the stored green coffee has not the correct moistures (10-12%) for storage. Moistures lower than 17% do not allow CBB reproduction. If you have the reference to support your statement, pleases include it or just delete it.
Reply: Thank you for your comment. We have revised it.
- Keywords: Add: Survival rate, fecundity, longevity. Delete: coffee berry borer, it was included in the title, so it does not need to be repeat.
Reply: We have corrected it.
- Line 50. Scientific names: include (Order and Family) for each of them.
Reply: We have added it in our manuscript.
- Line 61-62. Those statements are not necessarily true. Yes, there are few references, but you must to include those reference here. Look that nobody has conducted any biology or life history of CBB around of the world.
Reply: Thank you for your comment. We have cited the papers and correctd it in our manuscript.
- Line 63-73. All of this is relevant, but you only cite 3 references about the life history experiments and the methodology proposed “age-stage, two -sex life table”. However, here you must include some reference about CBB life history parameters previously conducted under field conditions or laboratory.
Reply: Thank you for providing us important literatures to refer to. We have cited those papers and revised our manuscript.
- You already have the last reference (Baker 1999), but please discuses in deep several aspects related to CBB life history and reproduction parameters that are included in this report.
Reply: We tried to revised it in the discussion part based on the work of Baker (1999).
- Line 87. The artificial diet … Brun et al. (1993). Why this old diet was used, knowing the novel more effective and improved diets are used to rearing CBB and its parasitoids. Reports from 2000 until 2022.
Reply: Thank you for your valuable information. It’s a pity that we found some new diet after our experiments has finished. Definitely we will try the new diets when we conduct new experiments in the future.
- Line 92. This first laboratory trial was a good experiment to determine the minimal impact of CBB survival caused by the methodology (observation intervals) during evaluations.
Reply: Thank you for your encouragement.
- Figure 1. Be specific: Hypothenemus hampei (Female). ?
Reply: We have revised it.
- Line 247. 27.64 – 59.85 days at 27 oC – 18 OC respectively.
Reply: We have corrected it.
- Figure 2. This figure 2 is not needed. All regressions are included on table 4. It is no necessary to repeat data or information.
Reply: We have deleted it in our manuscript.
- Line 373 and table 7. The mean generation time (T) for all different temperatures (18, 21, 24, and 27 oC ) looks too long for me. How do you explain that? Those data suggest that a constant temperature between 3 and 7 generation of CBB per year under laboratory conditions. Correct? From my experience rearing CBB and parasitoids we get about 1 generation of CBB each (38-45 days) at 24 o
Reply: Yes. it only takes 31.93 days under 24 oC. in our test (Table 2). We supposed what you mentioned may be mean generation time in the analyses of population parameters. It is different from a generation in the developmental stage.
- Line 531. Reference?
Reply: Thank you for your comment. This is our inference according to our results. We have revised our manuscript.
- In discussion: There are several factors that may affect the CBB reproduction under field conditions and laboratory conditions different of temperature. For example, the humidity is relevant for fecundity and developmental stages. Under field conductions and when berries are used in the lab for rearing CBB, the moisture of berry is critical to allow CBB reproduction. Finally, the use of diets is important to allow a continuous rearing of CBB in the lab a long of the year, which is needed for multiple uses. For example, for allow rearing of CBB parasitoids or conduct additional laboratory experiments for supply CBB adults for field trials as well. The age of developing berries is critical for CBB reproduction and the pre-oviposition time depends of it under field conditions. I understand that this work was conducted under laboratory conditions and it bring relevant information about all CBB life history parameters. However, in the discussion is important to include aspects about field conductions. For example, the number of CBB generations will dependent of many factors no only temperature. For example, flowering patter in some locations are different, which affect the distribution of berries along of the coffee season, and it affect the CBB reproduction and population as well. There is not the same planting coffee in Brazil, Colombia, Mexico, Hawaii, Tanzania, Kenya, India or Vietnam. Each location has its own weather conditions, which impact the coffee plant phenology and subsequently the CBB survival, reproduction ans population. So, those aspects need to be included in the discussion to see the implications of all your results not just for laboratory conductions and the potential of rearing CBB, but also, for rearing parasitoids or for facilitated better understanding CBB biology and its potential control (integrated pest management).
Reply: We have modified our manuscript.

Reviewer 2 Report
Authors have graphed and presented their results somewhat clearly, drawing attention to the implications of their findings. I found the study of interest and a good contribution to the knowledge of bio ecology of a coffee berry borer (CBB). The Insects journal is perhaps appropriate, but I suggest resubmitting the work once the following major corrections are made.
1) The Intro and Discussion provide no insight on how this MS relates to the various other ones cited in the text or concerns that have been raised by other researchers. The authors do not present any hypotheses or expectations that could be connected to previous studies; adding these details will improve the paper as indicated below in my comments. The authors should also clearly explain why the study was done, why it was important, and how it fits with other studies. It should be clear and concise. The intro should also include what outcome(s) they expect, and how it would help support or refute their hypotheses or answer their questions.
2) My primary concern is that the authors are extrapolating the applicability of their results beyond what the design supports. These are only data from a set of four highly artificial constant laboratory conditions (i.e., 18C-27C), so the inference power of the paper is very limited, but authors do not acknowledge this detail at all and need to be more forthcoming. The effect of fluctuating temperature profiles on CBB development, survival, and fecundity was not investigated in this study. This is a critical limitation of the study, and the authors must concede and discuss this. The interaction of cyclic temperatures with nonlinear characteristics of CBB development curves, for example, can introduce significant deviations from the results obtained in this study, and especially at the lower and higher temperatures of development functions which were not investigated at all (e.g., <18C and >27C). Studies across a broader set of fluctuating temperature regimes are therefore encouraged so that more realistic effect of temperature on biological parameters of CBB could be elucidated, as this is the closest to temperature fluctuations that occur in the field. So, I am suggesting to the authors to tone-down the language a little and admit that there are still substantive uncertainties to be considered.
3) Some of the authors’ statements would be much stronger if they tie their work to the body of literature that has built up on the bio ecology of insect parasitoids (e.g., see https://doi.org/10.1093/jee/toz067 and https://doi.org/10.1093/jee/toy429) and insect pests (see https://doi.org/10.1093/jee/toz320). These studies provide strong evidence that daily temperature fluctuations significantly affected development times and longevity of insects studied, resulting in marked deviations and potentially erroneous predictions when compared to their constant temperature regimen counterparts. In these studies, each fluctuating temperature profile was modeled after field recorded temperatures that had the desired average target temperature. These are the first studies ever to undergo such analysis. This article should provide details on all these fronts to provide the proper context for the work. This is not to diminish the data gathered in this study, as they are of value. But it is important for the authors not to overgeneralize, and to warn the reader, including regulatory agencies, against doing so as well. Adding these details will improve the discussion.
4) My other concern is that the estimated thresholds may not equate to the biological thresholds within which CBB develops as only being explained by the Lactin-2 model. Studies on the effects of rearing temperature on insect development have been criticized because analyses commonly use single models that are considered standard to the field of investigation or are preferred for a particular taxonomic group (see https://doi.org/10.1093/aesa/saw098, or https://doi.org/10.1093/aesa/sax063 for further explanation). As a result, alternative models that could provide superior fits to experimental datasets may be overlooked. Other important criteria that should be considered for fitting nonlinear models to temperature-driven development rate data for insects include use of parameters that have biological relevance are close-to-linear, which means that the least squares estimators are close to being normally distributed and are unbiased, minimum variance estimators. In these situations, good initial parameter estimates can be obtained which promotes successful model convergence (see https://doi.org/10.1093/jee/toz320 for further explanation).
5) Also, the discussion lacks real concluding remarks in my opinion, and if I was an IPM practitioner or consultant, I’d want to see these recommendations for my area or city.
Overall, I was excited to see the results of the paper after reading the abstract, but I found it hard to extract key messages useful to policymakers and professionals, probably in large part due to the lack of connection with other published work and need for improved structure of the current manuscript.
The next draft of this paper will need to be dramatically different to have a chance at publication in my opinion.
Author Response
Response to Reviewer 2 Comments
Thank you for your useful comments and professional suggestions. We tried to revise our manuscript according to your comments point by point.
- The Intro and Discussion provide no insight on how this MS relates to the various other ones cited in the text or concerns that have been raised by other researchers. The authors do not present any hypotheses or expectations that could be connected to previous studies; adding these details will improve the paper as indicated below in my comments. The authors should also clearly explain why the study was done, why it was important, and how it fits with other studies. It should be clear and concise. The intro should also include what outcome(s) they expect, and how it would help support or refute their hypotheses or answer their questions.
Reply: Thank you for your comment. We have revised it in our manuscript.
- My primary concern is that the authors are extrapolating the applicability of their results beyond what the design supports. These are only data from a set of four highly artificial constant laboratory conditions (i.e., 18C-27C), so the inference power of the paper is very limited, but authors do not acknowledge this detail at all and need to be more forthcoming. The effect of fluctuating temperature profiles on CBB development, survival, and fecundity was not investigated in this study. This is a critical limitation of the study, and the authors must concede and discuss this. The interaction of cyclic temperatures with nonlinear characteristics of CBB development curves, for example, can introduce significant deviations from the results obtained in this study, and especially at the lower and higher temperatures of development functions which were not investigated at all (e.g., <18C and >27C). Studies across a broader set of fluctuating temperature regimes are therefore encouraged so that more realistic effect of temperature on biological parameters of CBB could be elucidated, as this is the closest to temperature fluctuations that occur in the field. So, I am suggesting to the authors to tone-down the language a little and admit that there are still substantive uncertainties to be considered.
Reply: We have modified our sentences.
- Some of the authors’ statements would be much stronger if they tie their work to the body of literature that has built up on the bio ecology of insect parasitoids (e.g., see https://doi.org/10.1093/jee/toz067 and https://doi.org/10.1093/jee/toy429) and insect pests (see https://doi.org/10.1093/jee/toz320). These studies provide strong evidence that daily temperature fluctuations significantly affected development times and longevity of insects studied, resulting in marked deviations and potentially erroneous predictions when compared to their constant temperature regimen counterparts. In these studies, each fluctuating temperature profile was modeled after field recorded temperatures that had the desired average target temperature. These are the first studies ever to undergo such analysis. This article should provide details on all these fronts to provide the proper context for the work. This is not to diminish the data gathered in this study, as they are of value. But it is important for the authors not to overgeneralize, and to warn the reader, including regulatory agencies, against doing so as well. Adding these details will improve the discussion.
Reply: Thank you for providing us important comments and references. Yes. We did have significantly different results when we conducted analyses of constant temperature and variable temperature. We have tried to discuss more in our discussion and modified our manuscript.
4) My other concern is that the estimated thresholds may not equate to the biological thresholds within which CBB develops as only being explained by the Lactin-2 model. Studies on the effects of rearing temperature on insect development have been criticized because analyses commonly use single models that are considered standard to the field of investigation or are preferred for a particular taxonomic group (see https://doi.org/10.1093/aesa/saw098, or https://doi.org/10.1093/aesa/sax063 for further explanation). As a result, alternative models that could provide superior fits to experimental datasets may be overlooked. Other important criteria that should be considered for fitting nonlinear models to temperature-driven development rate data for insects include use of parameters that have biological relevance are close-to-linear, which means that the least squares estimators are close to being normally distributed and are unbiased, minimum variance estimators. In these situations, good initial parameter estimates can be obtained which promotes successful model convergence (see https://doi.org/10.1093/jee/toz320 for further explanation).
Reply: Thank you for your important suggestions and literature providing. After reading the papers you suggested, we totally agreed to your comments. Nonlinear analysis using the Lactin-2 model seems to cause serious bias in the result. We decided to delete the part using nonlinear analysis and revised our manuscript. We tried to discuss more in development and reproduction in our discussion part.
5) Also, the discussion lacks real concluding remarks in my opinion, and if I was an IPM practitioner or consultant, I’d want to see these recommendations for my area or city.
Reply: Thank you for your comments. We have discussed more in our manuscript.

Round 2
Reviewer 2 Report
Authors have done a nice job addressing my original comments and suggestions. Thank you.